# Evaluation of the Adverse Effects of Chronic Exposure to Donepezil (An Acetylcholinesterase Inhibitor) in Adult Zebrafish by Behavioral and Biochemical Assessments

**DOI:** 10.3390/biom10091340

**Published:** 2020-09-18

**Authors:** Gilbert Audira, Nguyen Thi Ngoc Anh, Bui Thi Ngoc Hieu, Nemi Malhotra, Petrus Siregar, Omar Villalobos, Oliver B. Villaflores, Tzong-Rong Ger, Jong-Chin Huang, Kelvin H.-C. Chen, Chung-Der Hsiao

**Affiliations:** 1Department of Chemistry, Chung Yuan Christian University, Chung-Li 320314, Taiwan; gilbertaudira@yahoo.com (G.A.); hieubtn90@gmail.com (B.T.N.H.); 2Department of Bioscience Technology, Chung Yuan Christian University, Chung-Li 320314, Taiwan; nguyen021194@gmail.com (N.T.N.A.); siregar.petrus27@gmail.com (P.S.); 3Department of Biomedical Engineering, Chung Yuan Christian University, Chung-Li 320314, Taiwan; nemi.malhotra@gmail.com (N.M.); sunbow@cycu.edu.tw (T.-R.G.); 4Department of Pharmacy, Faculty of Pharmacy, University of Santo Tomas, Manila 1008, Philippines; oavillalobos@ust.edu.ph; 5Department of Biochemistry, Faculty of Pharmacy, University of Santo Tomas, Manila 1008, Philippines; obvillaflores@ust.edu.ph; 6Center of Nanotechnology, Chung Yuan Christian University, Chung-Li 320314, Taiwan; 7Department of Applied Chemistry, National Pingtung University, Pingtung 900391, Taiwan; hjc@mail.nptu.edu.tw

**Keywords:** donepezil, acetylcholinesterase inhibitor, dementia, zebrafish, behavior

## Abstract

Donepezil (DPZ) is an acetylcholinesterase inhibitor used for the clinical treatment of mild cognitive impairment. However, DPZ has been reported to have adverse effects, including causing abnormal cardiac rhythm, insomnia, vomiting, and muscle cramps. However, the existence of these effects in subjects without Dementia is unknown. In this study, we use zebrafish to conduct a deeper analysis of the potential adverse effects of DPZ on the short-term memory and behaviors of normal zebrafish by performing multiple behavioral and biochemical assays. Adult zebrafish were exposed to 1 ppm and 2.5 ppm of DPZ. From the results, DPZ caused a slight improvement in the short-term memory of zebrafish and induced significant elevation in aggressiveness, while the novel tank and shoaling tests revealed anxiolytic-like behavior to be caused by DPZ. Furthermore, zebrafish circadian locomotor activity displayed a higher reduction of locomotion and abnormal movement orientation in both low- and high-dose groups, compared to the control group. Biomarker assays revealed that these alterations were associated with an elevation of oxytocin and a reduction of cortisol levels in the brain. Moreover, the significant increases in reactive oxygen species (ROS) and malondialdehyde (MDA) levels in muscle tissue suggest DPZ exposure induced muscle tissue oxidative stress and muscle weakness, which may underlie the locomotor activity impairment. In conclusion, we show, for the first time, that chronic waterborne exposure to DPZ can severely induce adverse effects on normal zebrafish in a dose-dependent manner. These unexpected adverse effects on behavioral alteration should be carefully addressed in future studies considering DPZ conducted on zebrafish or other animals.

## 1. Introduction

According to 2019 WHO guidelines on “Risk reduction of cognitive decline and dementia”, 10 million new cases of dementia are reported annually and this figure is set to triple by 2050 [1]. Dementia is a progressive neurodegenerative disease where brain cell death happens over time, leading to the loss of memory and executive function. Many pharmacological strategies, in the form of medicines such as donepezil (DPZ), rivastigmine, and galantamine, as well as therapies, have been devised to ameliorate the symptoms and, in some instances, slow the progression of dementia. These medications boost the levels of chemical messengers involved in memory and judgment, whereas therapies help to cope and to manage patient behavior.

DPZ, sold with the trade name of Aricept, is a reversible inhibitor of the enzyme acetylcholinesterase (AChE) and is one of the most widely used therapeutics. Inhibition of AChE helps to enhance cholinergic function by increasing the cortical amount of acetylcholine. It has been shown to be helpful in treating the symptoms of Alzheimer’s disease (AD) or dementia, including sleep disturbances, aggression, and reduction in memory, as well as slowing the progression of neuronal loss, advancing cognitive symptoms, and postponing cognitive degeneration in patients [2,3]. In younger and older humans, several studies have shown that the percentages of rapid eye movement (REM) sleep and REM density were enhanced, while REM latency was reduced by donepezil [4]. Another study in humans also found that donepezil-treated AD patients were less threatening, had a lower level of behavioral disturbances, and needed fewer sedatives than those not on donepezil [5]. Furthermore, a prior study in patients with mild–moderate AD discovered that donepezil administration improved mood, ability to perform the activities of daily living (ADL), disturbing behavior, and social behavior, as well as memory after 3 months. The behavioral changes were found to occur in a dose-dependent manner [6]. Nevertheless, it is already known that cholinesterase inhibitors exhibit some side-effects with increased cholinergic activity in AD patients, such as loss of appetite and gastrointestinal distress, including diarrhea, nausea, vomiting, muscle and abdominal cramps, and anorexia [7,8]. Unfortunately, these side-effects can prevent patients from achieving an effective dose of the drug [9].

In a prior study, the potential effects on neurodegeneration and cognitive deficits of DPZ were studied in rats. DPZ was administered to male Wistar rats 15 days before a 192-IgG-saporin injection, which can cause a secretive depletion of basal forebrain cholinergic neurons, which is a key component of the cognitive deficits associated with aging and dementia. Later, rats were subjected to several behavioral tests to analyze their anxiety levels, spatial working memory, sociability test, social motivation, and discriminative competencies, followed later by biochemical tests. They observed increased hippocampal and neocortical caspase-3 activity and impaired working memory, spatial discrimination, and social novelty preference without affecting anxiety levels and fear conditioning. Thus, the study concluded that pre-treatment of DPZ has beneficial effects on behavioral activity induced by cholinergic depletion and diminishing hippocampal and neocortical neurodegeneration [10]. Next, the neuroprotective effect of DPZ was evaluated on neurotoxicity induced by amyloid-beta (Aβ) (1-40) in primary cultures of rat septal neurons highly susceptible to Aβ toxicity. After Aβ (1-40) was added to the medium, DPZ was found to reduce lactate dehydrogenase (LDH) in a dose-dependent manner, suggesting that DPZ applies a neuroprotective effect by dramatically reducing the amount of toxin from Aβ fibrils in septal neuron cultures [11].

In current biomedicine research, zebrafish provide excellent experimental models to close the existing gap between in vitro and in vivo studies. As behavioral studies are important for a deeper understanding of the pathology and development of a neurological disease, zebrafish, as a model organism, provides an efficient alternative model platform for neurological and behavioral studies, as it shares similarities with the highly conserved nervous system structure and function of humans [12,13]. In a previous study in zebrafish, the effect of DPZ on zebrafish larvae was studied by vibrational startle response assay (VSRA). This assay was conducted to evaluate the habitation and escape response in zebrafish larvae. From the results, DPZ was found to be consistent with its acetylcholine receptor agonist role by significantly increasing the magnitude of startle response, area under the curve (AUC) value, and reduced habituation to VSR [14,15]. In another prior study, zebrafish larvae were also used for behavior analysis, with the help of a series of acoustic stimuli. Administration of DPZ increased acoustic startle response and reduced habituation in larvae, similar to a study using a rodent model [14].

It is already well-known that donepezil can improve cognitive dysfunction by enhancing cholinergic function in humans and several animal models (including zebrafish), even though various side-effects are still observed in human patients with Alzheimer’s disease. However, the effect of donepezil in enhancing cognition and intellectual function in normal-condition subjects remains unaddressed. Little is still known about its adverse effects on subjects without AD, especially in terms of behavior. In addition, considering its dose-dependent effects, a study to observe the different effects caused by the differences in DPZ concentration needs to be conducted. Thus, to further examine the positive and/or adverse effects of DPZ in subjects without neurodegenerative disorders and to further inspect its effects under different doses, we studied the effect of two concentrations of donepezil in normal adult zebrafish, particularly focused on their behavior. Moreover, to elucidate the possible mechanism that underlies its effect, biochemical assays were conducted. We hypothesized that chronic DPZ treatment would affect zebrafish behavior in the absence of neurodegeneration.

## 2. Experimental Methodology

### 2.1. Animal Husbandry

Wild-type AB strain zebrafish were raised and kept in standard laboratory conditions, according to the protocol described by Westerfield [16]. All procedures were approved by The Committee for Animal Experimentation of the Chung Yuan Christian University (Number: CYCU104024, Issue Date: 21 December 2015). Briefly, zebrafish were housed in a recirculating tank system on a 14:10 h light/dark cycle. A trapezoid tank with 34 cm at the top, 23 cm along the bottom, 19 cm along the diagonal side, 18 cm high, and 27 cm width filled with 8 L of water was used for holding the fish. Water quality was maintained at 28.5 °C (pH 7.2–7.6) with the conductivity of the system water kept between 300 and ~1500 µS. The circulating water was constantly filtered by ultraviolet (UV) light. All fish used in this study were fed twice daily (09:00 and 17:00) with fresh brine shrimp and commercial dry food. For all experiments, around 5–6-month old adult zebrafish were used. All behavioral tests were done between 10:00 and 16:00, except for the circadian locomotor activity test. 

### 2.2. Preparation and Exposure to Donepezil HCl

Donepezil hydrochloride (Donepezil HCl) was purchased from Sigma-Aldrich (St. Louis, MO, USA; CAT#D6821). Donepezil HCl was dissolved in 1% dimethyl sulfoxide (DMSO) to obtain a stock solution with a concentration of 1000 mg/L (ppm) and stored in the refrigerator (4–8 °C). For the first experiment, which was a T-maze test, zebrafish were randomly divided into 2 groups containing 18 animals each using the simple random allocation method to avoid experimental bias [17,18]. One of the groups served as control, while another group was exposed to 2.5 ppm of donepezil for 21 days. This initial experiment aimed to study whether donepezil at a relatively high concentration could improve the cognition of normal zebrafish or not. This particular concentration was used based on several prior studies, especially in rats, which found the positive effects of donepezil in a similar concentration on rat cognitive performance [19,20,21,22,23]. As no adverse or beneficial effects on cognitive performance in zebrafish were observed, the adverse effect of donepezil was further examined in other tests. In addition, another lower concentration was applied, in order to conduct a deeper study on its effect in zebrafish, especially on their behavior. In the following experiment, Zebrafish were randomly divided into 3 groups containing 30 animals each using the simple random allocation method [17,18]. In addition, this sample size determination was chosen as a statistical power analysis in the untreated group with a 90% confidence interval and a margin of error of 7 units showed that a sample of size 25 was needed [24]. The detail of *n* number calculation can be found in Appendix A. One of the groups served as control, while other groups were exposed to 1 ppm and 2.5 ppm of donepezil, respectively, for 21 days. Zebrafish behavior was observed at days 14 ± 1 (novel tank, mirror biting, predator avoidance, social interaction, and shoaling tests), 17 (color preference test), 19 (circadian locomotor activity test), and 21 (biochemical assay). All of the zebrafish in each group in every experiment were kept in a 4 L tank as, based on prior studies, this stocking density of zebrafish for the long-term helps to maintain low stress levels, with the assumption that all of the fish received equal amounts of the carrier [25,26,27,28]. Incubation water was changed every day and the donepezil concentration was maintained during incubation in all of the experiments. The schematic diagram of this experiment can be found in Appendix A.

### 2.3. T-maze Test (Short-Term Memory Test)

To observe the effects of donepezil on the memory of zebrafish, we used the T-maze test, referring to a previous publication reported by Ngoc Hieu et al. [29]. The T-Maze apparatus used in this study consisted of two deep-water arms and one straight long arm, which divided the inner space into a T-way intersection. The straight tunnel was separated with a white-color gate, dividing the start chamber and a novel arm, which branched into two short arms that led to deeper left/right chambers at a three-way junction. The apparatus included zones I, II (start chamber and novel arm), and III/IV (deep-water chambers) at each side of the T-way section. Based on the principle of the passive avoidance experiment [30], an electric stimulus was given in zone III (left arm) with mild electric shock (1–2 V, 0.3–0.5 mA.) In addition, for place preference conditioning in adult zebrafish [31], a green-cue was placed on one side of the wall in zone IV [32]. The camera for video recording (OPTO, CCD, China), NCH Debut Video software, and idTracker software [33] were used to observe the movement of the fish. First, the fish underwent habituation to minimize apparatus novelty stress [34]. In habitual group trials, 6 fish were placed in the starting chamber for 1 min before opening the gate, after which fish were allowed to explore the entire T-maze for 30 min. Before individual habituation trials, fish were placed into individual 600 mL plastic tanks to acclimate individually for 2 h. Afterward, each fish was placed in a starting chamber with its gate closed. After 30 s, the gate was opened and the fish was allowed to explore the T-maze for 5 min. Next, the fish was returned to the individual tank for 24 h prior to the start of training trials. For the training day, three training sessions were performed and a maximum of three electric shock times were given per session. In each session, fish were placed in a starting chamber for 30 s for acclimation. When fish swam into the left arm, they would immediately receive a mild electric shock. If it did not escape from the left arm in 20 s after receiving the electric shock, the fish would be retrained. Each training session ended when there was no entrance into the left arm within 5 min. Later, fish that completed three training sessions were stored in small plastic containers until the testing session, which consisted of the same protocol as the training trials, only without the electric shock. The testing session was based on time dependence, 24, 48, and 72 h after accomplishing the training trials. The latency of the first time fish crossed to the left arm in the habituation trial was recorded as the latency before training. The difference in the latency to enter the left arm of the apparatus (recorded for a maximum of 5 min) and the total number of electric shocks in training trials are important parameters for learning studies in zebrafish. Differences in the latency to enter the left arm between before training and testing were used as indices of memory retention. The difference in the time spent in the punish arm (left arm) and the time spent in the non-punish area between before training and testing were used as indices of the cognitive ability of zebrafish.

### 2.4. A Set of Behavioral Tests

Our battery of zebrafish behavioral tests consisted of novel tank, mirror biting, predator avoidance, social interaction, and shoaling tests. In these tests, a trapezoid test tank with 22 cm along the bottom, 28 cm at the top, 15.2 cm high, and 15.9 cm along the diagonal side was used during the experiment, which was filled with ~1.25 L of fish water, as described in the previous method [35]. Except for the novel tank test, after 5 to 10 min of fish acclimation in the tank, fish behaviors were recorded for a period of 5 min. In the novel tank test, video recording was started immediately after the fish was put into the test tank for one minute, repeated with a 5 min interval until 30 min had passed. In the mirror biting test, a mirror was placed on one side of the test tank to observe their behaviors of biting the mirror, which indicated their level of aggressiveness. The ‘biting’ was defined as the movement of the fish when they traced their reflection as they swim quickly up and down or back and forth and was measured by counting the duration when the fish stayed in front of the mirror and contacted the mirror. Meanwhile, in the predator avoidance and social interaction tests, a transparent separator was placed in the middle of the tank to separate the test fish from a convict cichlid (*Amatitlania nigrofasciata*) or their conspecific, respectively. The conspecific was placed to study the social behavior of the tested fish by observing the number of interactions between the tested fish and the conspecific, while the convict cichlid in the predator avoidance test was used to stimulate the fear level of the tested fish, such that they could display predator avoidance behavior. In the shoaling test, fish in groups of three were placed into each tank and their behavior regarding shoal formation was observed after acclimation. All zebrafish behaviors were processed by trained observers blinded to the treatments. The difference of *n* number between some behavioral tests happened due to the death of the zebrafish, which was caused by an unidentified reason. 

### 2.5. Circadian Locomotor Activity Test

The circadian locomotor activity test was conducted as described in our previous publication [36]. The dark/light cycle test apparatus consisted of six custom-made small fish tanks (20 × 10 × 5 cm) which were placed above a lightbox. For each tank, three fish were tested. For the light cycle, a light-emitting diode (LED) was used as a light source, while an infrared light-emitting diode (IR-LED) was used as a light source in the dark cycle. A 940 nm infrared camera with a magnifying lens was located above the experimental setup, in order to record the fish movements at 30 frames per second. The apparatus was placed inside an incubator to maintain the temperature. In this experiment, we recorded zebrafish locomotor activity for 1 min every hour. Several important behavioral endpoints—namely, average speed, average angular velocity, meandering, freezing, swimming, and rapid movement time ratios—were measured during the test.

### 2.6. Color Preferences Assay

The color preference assay was carried out in a 21 cm × 21 cm × 10 cm acrylic tank filled with 1.5 L of filtered water. Every tank consisted of four compartments, where each compartment was divided into two-color combinations among red, green, blue, and yellow colors. In total, there were six color combinations to determine the color preference of zebrafish. The color preference was recorded with a combination of an infrared (IR) camera and high-resolution HD camera and analyzed using idTracker software [33].

### 2.7. Estimation of Neurotransmitter Activity, Oxidative Stress Level, Antioxidant Enzyme Stress Hormone, and Neuropeptides in Brain Tissue

Six fish were randomly selected from each group for biochemical assays. For each group, six brains were divided into 3 samples (two brains per sample). Whole-brain tissue was homogenized on ice in 50 volumes (*v*/*w*) of PBS at pH 7.2 using a bullet blender tissue homogenizer (Next Advance, Inc., Troy, NY, USA). Later, samples were incubated on ice for 30 min before centrifuging at 12,000 g for 10 min. The crude homogenate was stored in 50 μL aliquots at −80 °C until their use in ELISA assays. Neurotransmitter levels of serotonin (5-HT), dopamine (DA), acetylcholine (ACh), acetylcholinesterase (AChE), melatonin, GABA (γ-aminobutyric acid), norepinephrine, oxytocin, and vasotocin were measured using target-specific ELISA kits purchased from a commercial company (ZGB-E1572, ZGB-E1573, ZGB-E1585, ZGB-E1637, ZGB-E1597, ZGB-E1574, ZGB-E1571, ZGB-E1668 and ZGB-E1673, Zgenebio Inc., Taipei, Taiwan). The levels of reactive oxygen species (ROS), superoxide dismutase (SOD), malondialdehyde (MDA), catalase (CAT), the stress hormone cortisol, and neuropeptide kisspeptin were measured using target-specific ELISA kits also purchased from a commercial company (ZGB-E1561, ZGB-E1604, ZGB-E1592, ZGB-E1598, ZGB-E1575, and ZGB-E1696, Zgenebio Inc., Taipei, Taiwan). The target protein content or activity was analyzed according to the manufacturer’s instructions. 

### 2.8. Determination of ATP, Oxidative Stress Level and Antioxidant Enzyme Activity in Muscle Tissues

Six fish were randomly selected from each group for biochemical assays. For each group, the muscle was dissected from two fish and put together as one sample. Similar to the brain samples, muscle samples were homogenized. Later, the levels of reactive oxygen species (ROS), superoxide dismutase (SOD), malondialdehyde (MDA), catalase (CAT), and ATP were measured using target-specific ELISA kits purchased from a commercial company (ZGB-E1561, ZGB-E1604, ZGB-E1592, ZGB-E1598, and ZGB-E1580, Zgenebio Inc., Taipei, Taiwan). The target protein content or activity was analyzed according to the manufacturer’s instructions. 

### 2.9. Statistical Analysis

All data are presented either as mean ± standard error of the mean (SEM) or median with interquartile range. Statistical analyses were conducted by either using a parametric test (*t*-test and ANOVA followed by least significant difference multiple comparison tests) or non-parametric tests (Mann–Whitney Test and Kruskal–Wallis tests followed by Dunn’s multiple comparisons test) by trained analysts blind to the experimental conditions. The non-parametric tests were applied as behavioral data generally do not follow a normal distribution [37]. However, to confirm their non-normal distribution, as the prior study mentioned, testing for normal distribution was still conducted in the data, as calculated by non-parametric tests prior to the calculation. Normality tests included Anderson–Darling, D’Agostino and Pearson, Shapiro–Wilk, and Kolmogorov–Smirnov tests. The Prism software (GraphPad Software version 7, La Jolla, CA, USA) was used to perform the statistical tests and output the graphs.

## 3. Results

### 3.1. Effect of DPZ Exposure in Short-Term Memory in Zebrafish

DPZ, as an acetylcholinesterase inhibitor, has been reported to improve cognitive impairment and global function in patients with mild to moderately severe Alzheimer’s disease [13,38]. First, we tested the impact of DPZ on short-term memory in healthy adult zebrafish using a T-maze test. A T-maze is an instrument that has been widely utilized to evaluate spatial learning and memory in rodents [39,40,41] and zebrafish [42,43] after exposure to chemical pollution [44,45] or pharmacological drugs [46,47,48]. Zebrafish learning and memory performance were tested after exposure to 2.5 ppm of DPZ for 21 days. The results showed that the DPZ-exposed group displayed no significant difference in both training (learning) latency and the total number of electric shocks received in training sessions, compared to the control fish, which indicated that DPZ exposure had no adverse effects on the learning process of adult zebrafish (*p* = 0.0832; F (1, 136) = 3.046; *p* = 0.1374; Figure 1A,B). Furthermore, freezing time also displayed no significant difference in before and after training between control and DPZ-exposed fish (*p* = 0.0488; F (1, 136) = 3.951; Figure 1C). On the testing day, DPZ-exposed fish showed no significant difference in terms of latency after training and time spent in the non-punish arm, compared to the control fish (*p* = 0.1656; F(1, 136) = 1.944; *p* = 0.0666; F (1, 136) = 3.42; Figure 1D,F). However, significantly reduced time spent in the punish-arm was observed in the DPZ-exposed fish in subsequent time points after the training, when compared to the control group (*p* = 0.0063; F (1, 135) = 7.695; Figure 1E). Taken together, these results demonstrate that chronic exposure to DPZ in normal adult zebrafish had no adverse or beneficial effects on their learning; however, a slight improvement, in terms of adult zebrafish short-term memory, was observed. As there was not any significant improvement in terms of cognitive performance displayed by the normal zebrafish after chronic exposure to 2.5 ppm DPZ, a lower concentration (1 ppm) of this compound was also used in the following tests to observe the effect of both concentrations in normal zebrafish, especially in their behavior.

### 3.2. Effects of DPZ in Novel Tank Test for Zebrafish

The novel tank test is a useful behavioral assay to measure fish anxiety when a fish is exposed to a novel environment. Normally, a zebrafish exhibits bottom-dwelling behavior when it is placed into a new environment and starts exploring the area after a few minutes of adaptation [49]. After chronic exposure for 14 days, DPZ-exposed zebrafish were observed for their locomotor activity and exploratory behavior in this test. From the results, it was found that zebrafish exposed to 1 ppm of DPZ showed no significant differences in average speed (*p* = 0.251; F(2, 87) = 1.404; Figure 2A) and freezing time movement ratio (*p* = 0.0857; F(2, 87) = 2.528; Figure 2B), compared to the control group, indicating that chronic exposure to DPZ in a low concentration had no significant effect on zebrafish locomotor activity during the test. In contrast, zebrafish treated with a higher concentration of DPZ (2.5 ppm) showed lower locomotor activity with a significantly high level of freezing time movement ratio (Figure 2B). Furthermore, DPZ-treated fish at both concentrations showed anxiolytic-like behavior based on significant alterations in all of the exploratory behavior endpoints after testing by Dunnett’s multiple comparisons test, which were higher time in the top duration (*p* = 0.0394; F(2, 87) = 3.358), the number of entries to the top (*p* = 0.0021; F(2, 87) = 6.631), total distance traveled in the top (*p* = 0.0702; F(2, 87) = 2.739), and lower latency to enter the top (*p* = 0.0025; F(2, 87) = 6.418) during the first 20 min of novel tank exposure (Figure 2C–F). The locomotion trajectories and behavioral changes in the novel tank test can be found in Figure 2J–L.

### 3.3. Effects of DPZ in Aggressive Behavior of Zebrafish

The mirror biting test is a well-established fish paradigm which is commonly used for studying the social/aggressive behavior of adult zebrafish [50,51,52]. Unaltered mirror biting behaviors of the DPZ-exposed zebrafish were observed in the high concentration group (*p* > 0.9999; *p* > 0.9999; Figure 3A,B). On the other hand, the results showed that low concentration DPZ exposure induced aggressiveness in zebrafish, as indicated by a significantly higher mirror biting time percentage and longest duration in the mirror side than the control group (*p* < 0.0001; *p* < 0.0001; Figure 3A,B). Furthermore, both concentrations of DPZ affected the locomotion of zebrafish, as indicated by the hypoactivity-like behavior displayed by treated fish during the test. DPZ exposure led to a significant reduction of average speed and swimming time movement ratio, while the freezing time movement ratio was significantly increased, compared to the control group (Appendix A). In addition, regarding the rapid movement time ratio, no significant differences were found between the treated groups and the control group (Appendix A). Taken together, these results indicate that DPZ led to a reduction of zebrafish locomotor activity and that a low dose of DPZ exposure induced a high aggressiveness level which was not observed in the high dose of DPZ. The locomotion trajectories and behavioral changes in the mirror biting test can be found in Figure 3C–E.

### 3.4. Effects of DPZ in Predator Avoidance Behavior of Zebrafish

Predator avoidance is an innate response of fish when they encounter their natural predators, in which they show high anxiety or even freezing behavior. In this test, we used convict cichlid (*Amatitlania nigrofasciata*) as a predator fish to induce the fear response of zebrafish. Six independent measurements in total were analyzed during the predator avoidance test, which were approaching predator time, average distance to the separator, average speed, freezing, swimming, and rapid movement time ratios. From the results, there was an indication that DPZ exposure did not alter the predator avoidance of zebrafish. This phenomenon was shown by no significant differences observed in approaching time percentage (*p* = 0.4207; *p* > 0.9999) and average distance to the separator (*p* = 0.4337; *p* = 0.9999) between treated and control groups (Figure 3F,G). Interestingly, while the locomotion activity of the high dose group was slightly decreased after 14 days of DPZ exposure, a low dose of DPZ showed more pronounced hypo-activity behavior during the test. This alteration was supported by a decrement in average speed and swimming time movement ratio and increment in freezing time movement ratio observed in the low concentration group (Appendix A). On the contrary, zebrafish exposed to 2.5 ppm DPZ only showed a reduction in rapid movement time ratio, while the other behavior endpoints were not found to be statistically different to the control group (Appendix A). These results indicated that the locomotion of zebrafish was reduced in a dose-dependent manner and DPZ exposure did not alter the predator avoidance response in adult zebrafish. The locomotion trajectories and behavioral changes in the predator avoidance test can be found in Figure 3H–J.

### 3.5. Effects of DPZ Exposure in Social Interaction of Zebrafish

The social interaction test is a useful model to study zebrafish social phenotypes. In this consideration, we used a conspecific social interaction test to investigate whether DPZ exposure had any deleterious effect on zebrafish social performance. Normally, zebrafish display strong sociability when placed in a tank with conspecific fish. Interestingly, the results showed that exposure to 1 ppm of DPZ increased zebrafish sociability, as indicated by a significant increase in interaction time percentage (*p* = 0.0152), longest duration at separator side (*p* = 0.0028), and shorter average distance to the separator (*p* = 0.0237), compared to the control group (Figure 4A–C). On another hand, these phenomena were not observed in the high concentration group (*p* = 0.3905; *p* = 0.7286; *p* = 0.3800; Figure 4A–C). In addition, the average speed of zebrafish in both DPZ treatment groups was not significantly different from that of the control group (*p* = 0.4711; *p* = 0.0900; Figure 4D). Overall, these results indicate that DPZ exposure induced a slight increment in zebrafish sociability in a dose-dependent manner. The locomotion trajectories and behavioral changes in the social interaction test can be found in Figure 4E–G.

### 3.6. Effects of DPZ Exposure in Shoaling Formation of Zebrafish

The shoaling test is another social behavior test to assess the interaction of an animal group, as they move together in co-ordinated movements in order to reduce anxiety and the risk of being captured by predators. Generally, adult zebrafish display a stable shoal cohesion. When assessing shoaling behavior, the distance reduction from one individual to another is the most significant behavioral endpoint to evaluate group cohesion. Therefore, measuring average inter-fish distance and average nearest/farthest neighbor distances provides an objective and reproducible of group cohesion measurement [53]. From the results, the high concentration DPZ-exposed group displayed loosened shoaling formation behavior. This alteration was demonstrated by a significantly higher level of average inter-fish distance (*p* = 0.0065) and average nearest (*p* = 0.0378) and farthest neighbor distances (*p* = 0.0046) of this treated group than the control fish (Figure 5D–F). However, these behavior endpoints were observed to be at a similar level in the low concentration and control groups (*p* > 0.9999; *p* = 0.5954; *p* = 0.8757; *p* = 0.3874). Furthermore, after DPZ exposure in both concentrations, the locomotor activity of exposed fish showed no significant difference from the control group (Appendix A). In addition, in agreement with other behavioral test results, DPZ also showed an anxiolytic effect in terms of zebrafish behavior. This phenomenon was supported by the high level of time in the top and average distance to the center of the tank (Appendix A). Finally, these results demonstrated the potential anxiety-reducing effects of DPZ in zebrafish. The locomotion trajectories and behavioral changes in the shoaling test can be found in Figure 5D–F.

### 3.7. Effects of Donepezil Exposure in Circadian Locomotor Activity of Zebrafish

In a human physiological study, an increase in sleep efficiency and shortening in sleep latency were observed after DPZ administration in patients with Alzheimer’s Type Dementia (ATD) [54]. Thus, it is intriguing to study the relationship between DPZ and circadian locomotor activity of zebrafish, which is closely related to its circadian rhythm, in the absence of neurodegeneration [55,56]. In this study, a circadian locomotor activity test was used to observe the chronic effects of DPZ exposure (19 days) on circadian locomotor activity patterns in normal adult zebrafish. Generally, as zebrafish is a typical diurnal fish species, it displays robust locomotion activity in the light cycle and sleep-like behavior in the dark cycle [36]. From the results, both treated groups displayed a decrement in their locomotor activity during the day and night cycles (Figure 6A), as indicated by the low level of average speed (*p* < 0.0001), swimming (*p* = 0.0394; *p* = 0.030; *p* < 0.0001; *p* < 0.0001), and rapid movement ratios (*p* < 0.0001), as well as the high level of freezing movement time ratio (*p* < 0.0001) (Figure 6B,E–H,K–M). In addition, regarding their movement orientation, the elevated level of meandering observed in both treated groups during light and dark cycles indicated a zigzag-like movement exhibited by the treated fish (*p* < 0.0001; Figure 6D,J). However, regarding their average angular velocity, both treated groups showed no significant difference to the control group in both cycles (*p* = 0.0565; *p* > 0.9999; *p* = 0.7076), except for a slight reduction in the 1 ppm group during the light cycle (*p* = 0.0139; Figure 6C,I). The P value from the main effects of each circadian locomotor activity endpoint can be found in Appendix A.

### 3.8. Effects of DPZ Exposure in Color Preference of Zebrafish

Zebrafish have many specific photoreceptors in their cone cells to distinguish colors. Any adverse chemical effects targeting zebrafish-specific photoreceptors or neurotransmitter expression may cause changes in their color preference. The color preference tracking technique has been applied to evaluate neurodegenerative disorders, as an index for pre-clinical appraisal of drug efficacy, and behavioral evaluation of toxicity. Moreover, color-based experiments have also been demonstrated to be associated with aversion, anxiety, or fear in the zebrafish [57]. Therefore, based on the well-known anxiolytic effect of DPZ, we suspected that DPZ might cause a slight difference in terms of zebrafish color preference. Thus, to verify this hypothesis, we performed a color preference test to investigate whether DPZ exposure affected zebrafish color preference. Generally, adult zebrafish have a clear color preference, ranked as follows: red > blue > green > yellow. Based on the results, both treatment groups did not show any significant changes, regarding their color preference ranking compared to the control group. However, the low concentration group displayed significant differences in the color choice index compared to the other groups, especially the control group. These differences were observed in the test with green–yellow, red–blue, red–yellow, and blue–yellow color combinations. In the green–yellow color combination, 1 ppm DPZ-exposed fish showed the lowest level of green color choice index among the three groups (*p* < 0.0001; F(5, 138) = 46.03; Figure 7B). Meanwhile, in the red–blue and red–yellow color combinations, 1 ppm DPZ-exposed fish showed a significantly higher choice index for red color than the other groups (*p* < 0.0001; F(5, 138) = 189.3; *p* < 0.0001; F(5, 138) = 480.2; Figure 7D,E). Furthermore, the most interesting results were observed in the blue–yellow color combination. In this color combination, even though the low concentration group showed a lower blue color choice index than the control, the high concentration group displayed the lowest choice index of blue color among these three groups (*p* < 0.0001; F(5, 138) = 302.3; Figure 7F). However, in the green–blue and green–red color combination groups, both DPZ-exposed groups displayed similar blue and red color choice indices, respectively, to control fish (*p* < 0.0001; F(5, 138) = 24.09; *p* < 0.0001; F(5, 138) = 80.20; Figure 7A,C). Taken together, zebrafish exposed to 2.5 ppm of DPZ did not show any significant difference regarding their color choice index in all of the color combinations, except in the blue–yellow combination (with a significantly low level of the blue color choice index), while 1 ppm of DPZ changed zebrafish color choice indices in several color combinations (Figure 7F).

### 3.9. Effect of DPZ Exposure on Biomarker Expression in Zebrafish

In previous behavioral tests, DPZ exposure has been proven to induce adult zebrafish behavioral alterations in exploratory behavior, aggressiveness, social interaction, shoaling formation, and circadian locomotor activity, where these alterations might indicate an anxiolytic-like behavior. We examined the expression of biomarkers, in order to gain insight into the molecular mechanisms underlying the observed behavioral changes caused by DPZ exposure and to investigate the possible corresponding mechanisms controlling these behavioral alterations, by performing biochemical assays on some important biomarkers (Table 1). First, we tested the hypothesis that DPZ treatment reduces oxidative stress by evaluating several biomarkers of oxidative pathways in the brain and muscle by performing ELISA (Enzyme-linked immunosorbent assay) with target-specific antibodies. In brain tissues, DPZ exposure had no significant influence on reactive oxygen species (ROS), superoxide dismutase (SOD), and catalase (CAT) expression in the brain. However, DPZ treatment was found to elevate the lipid oxidation marker of the malondialdehyde (MDA) level (*p* = 0.005). In muscle tissues, DPZ exposure induced more robust biomarker changes, as shown by an elevation in ROS (*p* = 0.0478) and MDA (*p* = 0.0086) contents. Next, the determination of neurotransmitters was also performed in the brain tissues, in order to examine the possible mechanism underlying the alterations in behavioral responses of zebrafish after DPZ chronic exposure. By ELISA, the relative amounts of neurotransmitters or hormones such as cortisol, acetylcholine (Ach), acetylcholine esterase (AChE), serotonin (5-HT), norepinephrine, dopamine, γ-aminobutyric acid (GABA), kisspeptin, oxytocin, and vasotocin in the brain tissues of zebrafish could be precisely quantified and compared. From the results, it was shown that chronic DPZ exposure reduced the relative content of cortisol (*p* = 0.0193), while the oxytocin content was elevated in the brain (*p* = 0.0184). On another hand, other neurotransmitters or hormones displayed no significant differences when compared to the control group.

## 4. Discussion

Based on the findings regarding the side-effects of DPZ, regardless of its tolerability and evidence of its benefits on cognitive function, in the present study, we aimed to test the hypothesis that chronic exposure to DPZ leads to changes in behavior in healthy individuals using zebrafish as a model, as well as exploring the underlying biochemical processes affected. From the results, 2.5 ppm DPZ-exposed fish with no neurodegenerative disorder did not show any significant difference to control fish, in terms of cognitive performance. Interestingly, exposure to DPZ produced a slight improvement in the short-term memory of adult zebrafish, through the increased retention of adverse stimuli, as has been displayed in earlier studies on Alzheimer’s disease patients [58,59]. It is already well-known that the efficacy of DPZ in cognitive improvement is related to the increase in ACh levels in the central nervous system through inhibiting AChE activity [60,61,62]. In addition, oxidative stress is one of the major factors contributing to neuronal degeneration in the brain implicated in neurobehavioral disorders [63,64,65]. As the main antioxidant enzymes, SOD and CAT play important roles in the defense system against oxidative stress [66]. In previous studies in mice, DPZ has been demonstrated to act as an antioxidant, an important aspect for neuroprotection, indicating its effective role in memory enhancement by decreasing biochemical markers of oxidative stress level, such as MDA [67,68]. Taken together, the assessment of ACh and AChE activities, as well as antioxidant enzyme level in the brain of zebrafish, is needed to understand the underlying mechanisms of the cognitive enhancement role of DPZ in normal zebrafish. In the present study, even though they did not reach a statistically significant level, ACh, and AChE levels tended to be slightly increased in the DPZ-treated groups, compared to the control group. Furthermore, we also found that the MDA level was significantly reduced in the brain of zebrafish exposed to 2.5 ppm DPZ, when compared to control fish. Thus, it was indicated that even the levels of ACh, AChE, and oxidative stress activities were not significantly altered after chronic exposure to donepezil; this condition is sufficient to enhance the retention of adverse stimuli in normal zebrafish, despite no significant difference occurring in their cognitive performance. 

Next, DPZ was found to reduce zebrafish locomotor activity in a dose-dependent manner. This phenomenon was observed in most of our behavior assessments, including novel tank, mirror biting, predator avoidance, and circadian locomotor activity tests. Mostly, zebrafish displayed a higher reduction and variation of locomotor activity when exposed to the higher dose (2.5 ppm) of DPZ. This phenomenon may be related to the adverse effect of cholinesterase inhibitor drugs, as muscle cramps, weakness, and motor impairment are symptoms frequently associated with these drugs [69,70]. 

Another important finding in this study was a significant increase in the aggressiveness of zebrafish after exposure to low-dose donepezil. This finding was consistent with a previous publication in humans, reported by the Medicines Control Agency of the U.K., showing 41 cases of aggression or agitation in a total of 695 reports of a possible adverse drug reaction to DPZ [71]. Some patients became very paranoid and exhibited violent behaviors and, according to the manufacturer’s report, 5% of patients became agitated (although only 1% of which showed physical aggression) after donepezil admission [72]. Next, during the novel tank and shoaling tests, anxiolytic-like behavior was observed in the DPZ-treated fish. In the novel tank test, this behavior was indicated by the longer duration spent in the top portion of the tank displayed by the DPZ-exposed fish, compared to the untreated fish. Normally, as displayed by the control group, zebrafish typically engage in the bottom area of the tank when first moved into a novel environment and start to explore the top area after a few minutes of acclimation. Therefore, more time spent in the upper part of the tank, as well as higher frequency to enter the upper area of the tank, indicate anxiolytic-like behavior in adult zebrafish [49,73]. Furthermore, loosened shoal formation in the treated group might also indicate an anxiolytic-like behavior, as previous studies have shown that anxious fish tend to swim closer together in tighter shoals, compared to non-anxious fish, when placed in a novel environment [73,74]. Taken together, even though the reduction in anxiety in humans may be the result of improvement in executive function, these results are in line with recent research that discovered the positive contribution of DPZ in moderation of anxiety in patients with mild neurocognitive impairment (MCI), where anxiety symptoms were alleviated after 2 weeks of treatment with DPZ [75]. When studying anxiety, stress hormone levels can be used as a physiological endpoint to be paralleled with behavioral responses [76]. In this study, the biochemical assay showed decreased cortisol levels as well as slightly increased CAT levels in DPZ-exposed fish, compared to the control group. In zebrafish, cortisol is one of the physiological phenotypes which is used as a primary stress response hormone relevant to human stress physiology [77]. Excessive increases in cortisol level have been associated with hippocampal neuron dysfunction, as well as causing depression [78,79,80]. The alteration of stress hormone levels can also be valuable additions to parallel with behavioral observations in the study of anxiety [76]. Consistent with our findings, previous research demonstrated that chronic exposure to ethanol led to increased time spent in the upper zone with a trend towards reduced cortisol levels, indicating anxiolytic behavior in zebrafish [49]. In addition, therapies which can reduce cortisol levels also have a positive effect in treating Alzheimer’s disease (AD)-or depression-related symptoms, namely, stress, anxiety, and cognitive impairment [81,82,83]. Meanwhile, regarding the catalase result, a prior study discovered that CAT over-expression reduced anxiety in mice conducting the zero maze task [84]. Additionally, improved depression-like behavior associated with Alzheimer’s disease was shown to be dependent on CAT level increase induced by DPZ administration [85]. 

The social interaction test, another social interaction assay, showed a slightly increase in conspecific social interaction frequency in the low concentration DPZ group. In line with this result, disrupted shoaling was also displayed by the treated fish in the high concentration group. This set of results indicates that DPZ had noticeable effects on several measures used to determine zebrafish social behavior. Furthermore, these findings have been supported by a prior study which found that scopolamine-induced social behavior impairment in rats was clearly reversed by DPZ treatment [86]. Furthermore, another study found that patients with mild to moderate AD had improved social interaction, engagement, and interest after receiving treatment with DPZ [86,87,88].

Circadian rhythms of locomotor activity in zebrafish have been analyzed both in larvae and in adult fish, and are useful for studying the physiology of the adult zebrafish circadian system and the components of the system which are important for the generation and regulation of circadian rhythms of locomotor activity [89,90]. In addition, even though zebrafish locomotor activity was also measured in this test (as in the novel tank test), there was a fundamental difference that differentiated these tests from each other. In this test, the tested fish locomotor activity was measured after it was already acclimated in the environment while, in the novel tank test, its locomotor activity was observed when it was exposed to the new environment directly. From the results, it was shown that both concentrations of DPZ affected circadian locomotor activity, including an abnormal zigzag-like movement observed during both light and dark cycles. These abnormalities may be associated with central nervous system events affected by cholinesterase inhibition, such as sleep disorder; this symptom has mostly been reported in subjects treated with DPZ [91,92]. It has been shown that melatonin is an attractive candidate to mediate circadian processes, as the clock regulates its production (i.e., high-level signal during the night and low-level signal during the day) [93]. Moreover, melatonin has been reported to promote a sleep-like state in zebrafish [94] and to play a role in potentially linking circadian and homeostatic control of sleep [95]. In addition, cortisol has also demonstrated an important role in organizing the circadian system [80]. Cortisol peaks during daytime and an unusual increase in cortisol levels can be considered as risk factors of disease development [96]. Furthermore, even partial acute sleep loss can induce a higher cortisol level, compared to a normal sleep schedule [97]. In this study, based on the measured biochemical endpoints, we found a relationship between reduction of cortisol level and the abnormal circadian locomotor activity behavior of zebrafish after DPZ exposure. This result indicated that there was a possibility of DPZ having an effect in sleep disturbance, even in the absence of melatonin alteration. In a prior study, the potential effect of DPZ after its administration was discovered, as sleep efficiency was increased with shortened sleep latency in patients with Alzheimer’s Type Dementia (ATD) [54], having an adverse effect in a patient with night-time disturbance (NTD); thus, more research about the time of administration of DPZ is necessary [98]. However, one has to keep in mind that, even though the daily rhythms of locomotor activity in adults and larvae zebrafish can be attributed to the circadian system, there are other ways to evaluate circadian rhythms in zebrafish besides locomotor activity, namely, hormone levels and sleep behaviors, such as sleep latency or sleep bouts/efficiency. These measurements are worth trying, in a future study, to confirm the effects of DPZ on sleep observed in the current study [55,56].

Next, the color preference test did not show any significant change in color discrimination sequence (ranking) in both concentrations of DPZ. However, in different color combinations, it was observed that treated zebrafish displayed the highest preference intensity for red color, which aligned with an earlier study which indicated the preference of zebrafish for certain colors (e.g., red) over other colors when tested in the place preference (PP) box and T-maze [99]. 

Beside cholinergic system analysis, other neurotransmitter assays are also necessary to validate the underlying mechanism in behavior response of zebrafish after donepezil exposure, as neurotransmitter modulation has been associated with all practically important physiological systems in the brain; thus, neurotransmitter alteration may affect several physiological behaviors that are controlled by them [100]. Dopamine and serotonin are two key neurotransmitters which regulate brain function and behavioral responses [101]. The relation of dopamine to locomotion, cognition, and even emotion has been reported before [102]; whereas serotonin has been demonstrated to be associated with aggressiveness, anxiety, as well as social interaction behavior [102,103]. Unfortunately, the relationship between unaltered dopamine level and decreased locomotor activity from donepezil exposure in zebrafish remains unknown; thus, further studies are needed to investigate the other effect linkages. However, we suggest that the decrease in locomotor activity was caused by muscle damage, considering the high level of MDA, as the status of this compound can provide an estimate of muscle stress [104]. Oxidative stress in muscle tissue was also indicated by the distinct production of ROS in the muscle of high concentration DPZ-exposed fish. In addition, the GABA receptor has been reported to modulate anxiety response, while norepinephrine (NE) plays an important role in mediating the responses related to fear and stress [105]. However, serotonin, GABA, and NE were not altered after donepezil exposure, suggesting the higher level of aggressiveness and anxiolytic behavior displayed by the exposed fish was not related to the alteration of these neurotransmitters. Furthermore, recent studies have also reported that the signaling of several neuropeptides, such as kisspeptin, is associated with some behavioral responses, including fear, anxiety, and mood [106]. In this study, the kisspeptin level of donepezil-exposed fish showed no significant difference, which may validate the similar predator avoidance behavior between treated fish and control fish. On the other hand, vasotocin, a nonapeptide hormone, has been demonstrated to reduce social interaction and shoaling behavior in zebrafish [107]. However, in this study, we could not find any related vasotocin alteration in social interaction and shoaling behavior after donepezil exposure. Nevertheless, oxytocin, another nonapeptide hormone, was significantly increased after exposure to 1 ppm donepezil for 14 days. This phenomenon might be related to the increased social interaction and increased aggressiveness of 1 ppm DPZ-exposed fish, as oxytocin has been demonstrated to reverse social interaction and aggression deficits in adult zebrafish induced by dizocilpine (MK-801) [108]. Moreover, it also acts as a crucial regulator in social interaction and the aggressive behavior of mammals [107]. Finally, it is intriguing that the chronic exposure effect of donepezil observed in the current experiment was different from the acute exposure effect of DPZ observed in the previous study by Giacomini et al., where 24 h incubation with DPZ in adult zebrafish yielded anxiogenic-like effects which were paralleled by analyses of AChE activity in the brain [109]. This difference is reasonable, as another prior study found that this drug induced changes in a number of neurotransmitters, such as dopamine, serotonin, and norepinephrine, in some specific regions including the rat dorsal hippocampus, in a time-dependent manner [110]. In addition, other studies are in progress to determine the pharmacological effects associated with acute treatment with this compound.

## 5. Conclusions

In summary, consistent with our hypothesis, the current study demonstrated the capability of donepezil in causing behavioral alterations in normal zebrafish; moreover, the outcomes showed significant variation with the dose of donepezil treatment, which is consistent with previous clinical studies (the overall results are summarized in Figure 8). The chemical composition of a substance plays a significant role in its toxicity mechanism inside an organism; therefore, the dissolution of concentration in the soluble form inside an organism must be understood specifically, in terms of particle-mediated toxicity. The side-effects of donepezil usually decrease after a few days or may be relieved by the maintenance of the present dose level, by omitting one or more doses, or by temporarily decreasing dosage [92,111]. In addition, even though donepezil treatment did not cause any significant difference in zebrafish cognitive performance, a slight improvement in adult zebrafish short-term memory was observed in the treated fish. However, the mechanisms underlying donepezil-associated responses in zebrafish behaviors still require further clarification. Therefore, further investigations of potential neuronal pathways are needed to understand more precisely the mechanisms underlying the behavioral responses observed.

## Figures and Tables

**Figure 1 biomolecules-10-01340-f001:**
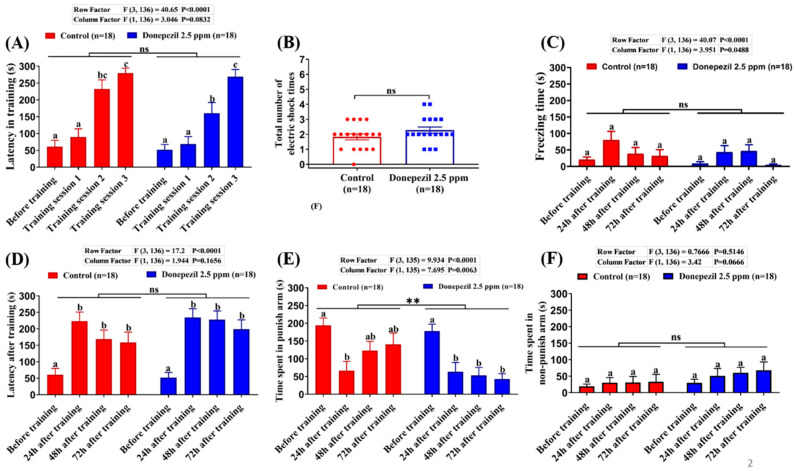
T-maze conditioning passive avoidance to test short-term memory of fish after 21 days exposure to 2.5 ppm of donepezil: (**A**) the latency of control and donepezil-exposed fish swimming into the punish chamber for each training session; (**B**) the total number of electric shocks given for successful training between control and donepezil-exposed fish; (**C**) the freezing time of control and donepezil-exposed fish at different time points before and after training; (**D**) the memory retention of control and donepezil-exposed fish at different time points before and after training; (**E**) the time spent in punish arm of control and donepezil-exposed fish at different time points before and after training; (**F**) the time spent in the non-punish area of control and donepezil-exposed fish at different time points before and after training. The data are expressed as the mean ± standard error of the mean (SEM) values. Different letters (a, b, c) on the error bars represent the significant differences (*p* < 0.05); A and C–F were analyzed by two-way ANOVA with Tukey-HSD (Honestly Significant Difference) post hoc test; B was analyzed by unpaired t-test; *n* = 18, ** *p* < 0.01.

**Figure 2 biomolecules-10-01340-f002:**
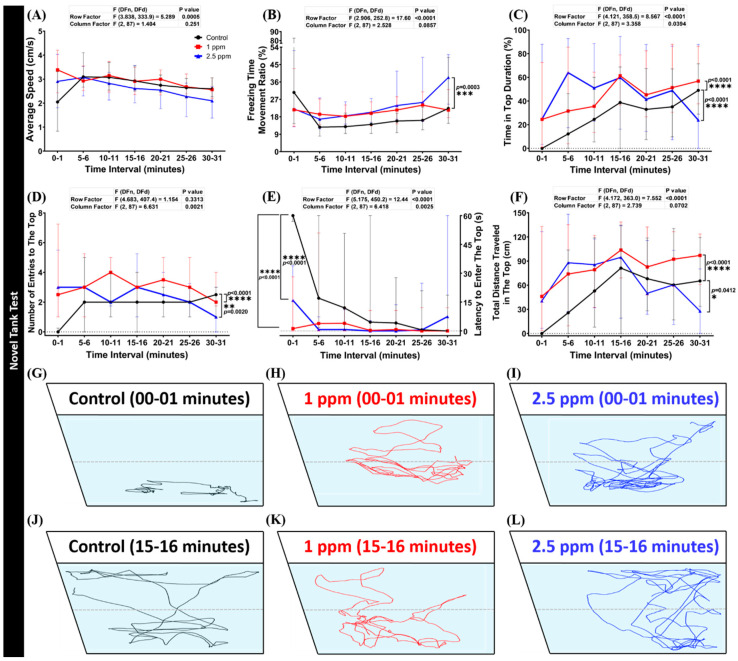
Behavior endpoints of control and donepezil-exposed zebrafish in novel tank test after 14 days of incubation: (**A**) average speed; (**B**) freezing time movement ratio; (**C**) time in top duration; (**D**) number of entries to the top; (**E**) latency to enter the top; and (**F**) total distance traveled in the top were analyzed. The swimming trajectories of (**G**,**J**) control; (**H**,**K**) 1 ppm Donepezil-exposed fish; and (**I**,**L**) 2.5 ppm Donepezil-exposed fish for novel tank test after 1 min and 15 min were recorded and compared. The black line represents control and the red line represented 1 ppm Donepezil-exposed fish, while the blue line represents 2.5 ppm Donepezil-exposed fish. The data are expressed as the median with interquartile range and were analyzed using a two-way repeated measure ANOVA with Geisser–Greenhouse correction. To observe the main column (donepezil) effect, Dunnett’s multiple comparison test was carried out; *n* = 30, * *p* < 0.05, ** *p* < 0.01, *** *p* < 0.005, **** *p* < 0.001.

**Figure 3 biomolecules-10-01340-f003:**
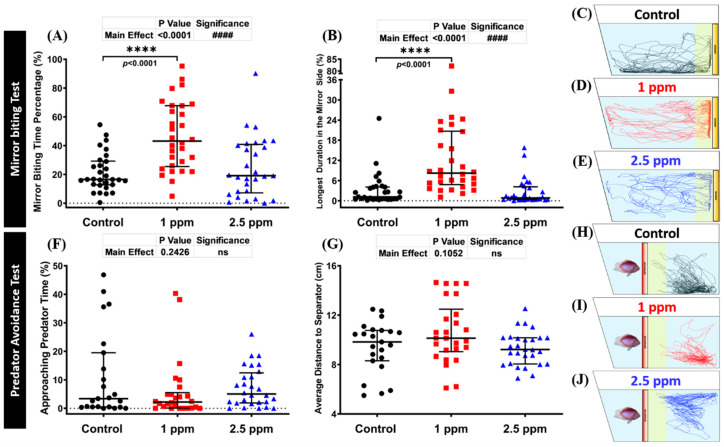
Mirror biting and predator avoidance behavior endpoint comparisons between the control, 1 ppm, and 2.5 ppm donepezil-exposed zebrafish groups after 14 days of exposure: (**A**) mirror biting time percentage and (**B**) longest duration in the mirror side were analyzed in the mirror biting. The swimming trajectories of (**C**) control, (**D**) 1 ppm Donepezil-exposed fish, and (**E**) 2.5 ppm Donepezil-exposed fish for the mirror biting test were recorded and compared (*n* = 30 for control and 1 ppm groups, *n* = 29 for 2.5 ppm group). (**F**) Approaching predator time and (**G**) average distance to the separator were measured in the predator avoidance test. The swimming trajectories of (**H**) control, (**I**) 1 ppm Donepezil-exposed fish, and (**J**) 2.5 ppm Donepezil-exposed fish for predator avoidance test were recorded and compared (*n* = 23 for the control group, *n* = 26 for 1 ppm group, *n* = 28 for 2.5 ppm group). The data are expressed as the median with interquartile range and were analyzed using the Kruskal–Wallis test followed by Dunn’s multiple comparisons test (####/**** *p* < 0.001, ns = not significant).

**Figure 4 biomolecules-10-01340-f004:**
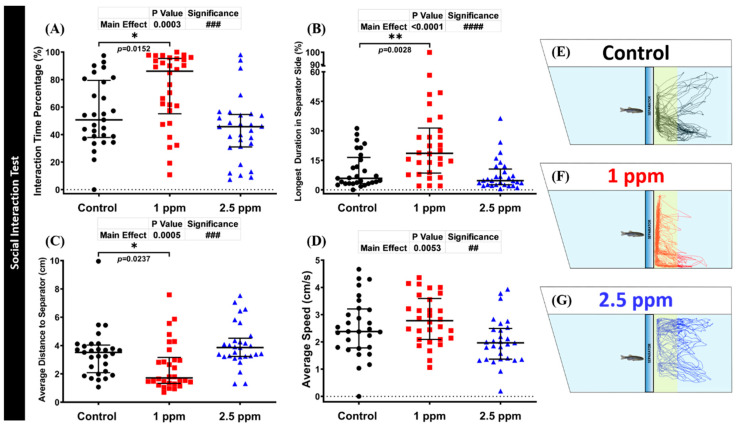
Social interaction behavior endpoint comparisons between the control group, 1 ppm, and 2.5 ppm donepezil-exposed zebrafish groups after 14 days of exposure: (**A**) interaction time percentage; (**B**) longest duration at separator side; (**C**) average distance to the separator; and (**D**) average speed were analyzed. The swimming trajectories of (**E**) control, (**F**) 1 ppm Donepezil-exposed fish, and (**G**) 2.5 ppm Donepezil-exposed fish for social interaction tests were recorded and compared. The data are expressed as the median with interquartile range and were analyzed using the Kruskal–Wallis test followed by Dunn’s multiple comparisons test (*n* = 29 for control and 2.5 ppm groups, *n* = 30 for 1 ppm group, * *p* < 0.05, ##/** *p* < 0.01, ### *p* < 0.001, #### *p* < 0.0001).

**Figure 5 biomolecules-10-01340-f005:**
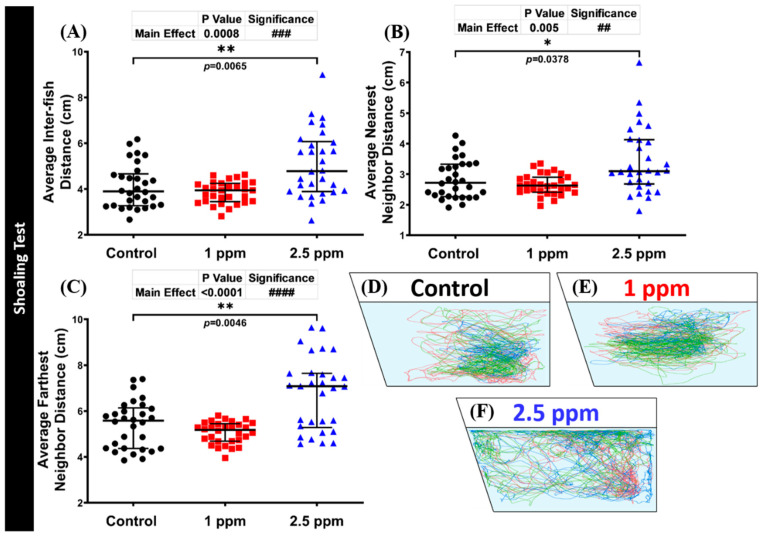
Shoaling behavior endpoint comparisons between the control group, 1 ppm, and 2.5 ppm donepezil-exposed zebrafish groups after 14 days of exposure: (**A**) average inter-fish distance; (**B**) average nearest neighbor distance; and (**C**) average farthest neighbor distance were analyzed. The swimming trajectories of (**D**) control, (**E**) 1 ppm Donepezil-exposed fish, and (**F**) 2.5 ppm Donepezil-exposed fish for shoaling tests were recorded and compared. Groups of three fish were tested for shoaling behavior. The data are expressed as the median with interquartile range and were analyzed using the Kruskal–Wallis test followed by Dunn’s multiple comparisons test (*n* = 30, * *p* < 0.05, ##/** *p* < 0.01, ### *p* < 0.001, #### *p* < 0.0001).

**Figure 6 biomolecules-10-01340-f006:**
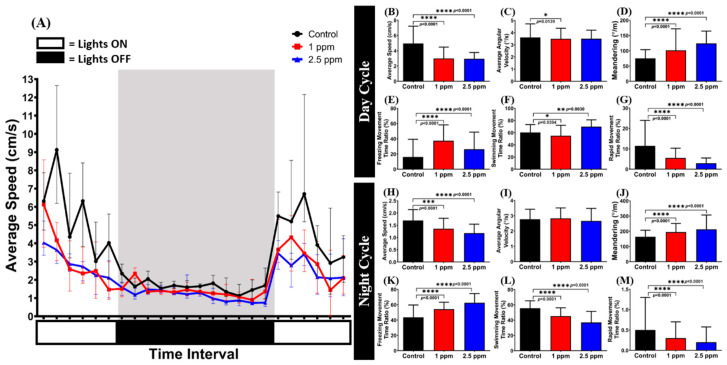
Evaluation of the circadian locomotor activity for control and chronic donepezil-exposed fish: (**A**) Circadian patterns of average speed; (**B**) average speed; (**C**) average angular velocity; (**D**) meandering; (**E**) freezing movement time ratio; (**F**) swimming movement time ratio; and (**G**) rapid movement ratio during the light cycle. (**H**) Average speed; (**I**) average angular velocity; (**J**) meandering; (**K**) freezing movement time ratio; (**L**) swimming movement time ratio; and (**M**) rapid movement ratio during the dark cycle. The data are expressed as the median with interquartile range and were analyzed using Kruskal–Wallis test followed by Dunn’s multiple comparisons test (*n* control fish = 18; *n* donepezil-treated fish = 18; * *p* < 0.05, ** *p* < 0.01, *** *p* < 0.001, **** *p* < 0.0001).

**Figure 7 biomolecules-10-01340-f007:**
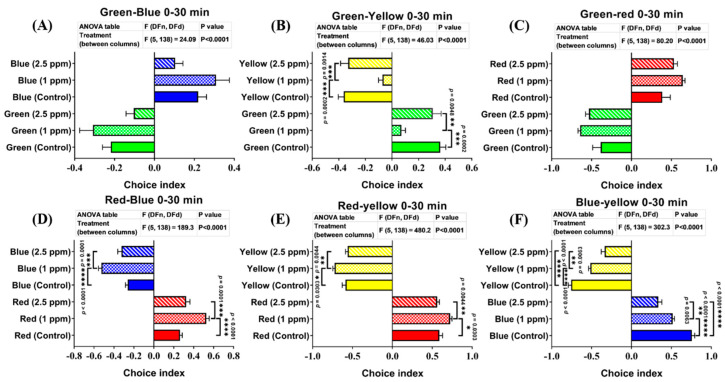
Comparison of the color preference behavior between control and donepezil-exposed fish after 17 days of donepezil exposure. The combinations of four colors were: (**A**) the green/blue test, (**B**) green/yellow test, (**C**) green/red test, (**D**) red/blue test, (**E**) red/yellow test, and (**F**) blue/yellow test. The data are expressed as the mean ± SEM values and were analyzed by two-way ANOVA (*n* = 24; * *p* < 0.05, ** *p* < 0.01, *** *p* < 0.001, and **** *p* < 0.0001).

**Figure 8 biomolecules-10-01340-f008:**
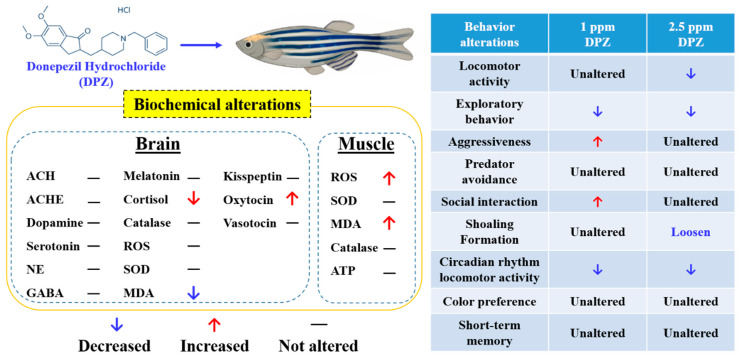
Summary of biochemical and behavioral changes in adult zebrafish after acute or chronic exposure to Donepezil. The behavioral alterations are summarized in the right panel and the biochemical alterations in brain and muscle tissues are summarized in the left panel.

**Table 1 biomolecules-10-01340-t001:** Comparison of biochemical levels in brain and muscle tissue between control fish and donepezil-exposed fish by enzyme-linked immunosorbent assay (ELISA). Data are expressed as the mean ± SEM values and were analyzed by one-way ANOVA (*n* = 3; * *p* < 0.05, ** *p* < 0.01).

Biomarker	Control	1 ppm	2.5 ppm	Unit	Significance	*p*-Value
**Brain**						
Reactive Oxygen Species (ROS)	1.478 ± 0.220	1.696 ± 0.133	1.437 ± 0.108	U/μg of total protein	NO	0.812
Superoxide Dismutase (SOD)	0.324 ± 0.023	0.315 ± 0.028	0.294 ± 0.013	U/μg of total protein	NO	0.6533
Malondialdehyde (MDA)	0.010 ± 0.0003	0.009 ± 0.0004	0.008 ± 0.0002 (**)	nmol/μg of total protein	YES	0.0055
Cortisol	2.900 ± 0.190	2.485 ± 0.081	2.021 ± 0.168 (*)	pg/μg of total protein	YES	0.0193
Acetylcholine (ACh)	0.0315 ± 0.003	0.0398 ± 0.002	0.0372 ± 0.002	ng/μg of total protein	NO	0.0936
Acetylcholine esterase (AChE)	0.173 ± 0.021	0.183 ± 0.014	0.183 ± 0.005	nmol/μg of total protein	NO	0.5106
Serotonin (5-HT)	0.363 ± 0.012	0.372 ± 0.012	0.334 ± 0.008	ng/μg of total protein	NO	0.1437
Norepinephrine	0.011 ± 0.0006	0.010 ± 0.0004	0.011 ± 0.0006	ng/μg of total protein	NO	0.4131
Dopamine	0.175 ± 0.001	0.210 ± 0.011	0.199 ± 0.008	U/μg of total protein	NO	0.0716
γ-aminobutyric acid (GABA)	1.05 × 10^−5^ ± 1.7 × 10^−7^	1.09 × 10^−5^ ± 6.6 × 10^−7^	9.90 × 10^−6^ ± 4.9 × 10^−7^	μmol/μg of total protein	NO	0.4104
Melatonin	0.129 ± 0.005	0.133 ± 0.003	0.130 ± 0.009	pg/μg of total protein	NO	0.8793
Kisspeptin	0.798 ± 0.043	0.736 ± 0.010	0.746 ± 0.034	ng/μg of total protein	NO	0.4026
Oxytocin	0.045 ± 0.001	0.060 ± 0.002 (*)	0.048 ± 0.004	pg/μg of total protein	YES	0.0184
Vasotocin	2.603 ± 0.118	2.234 ± 0.146	2.216 ± 0.056	pg/μg of total protein	NO	0.0891
**Muscle**						
ROS	1.368 ± 0.141	1.479 ± 0.096	1.893 ± 0.120 (*)	U/μg of total protein	YES	0.0478
SOD	0.241 ± 0.022	0.237 ± 0.013	0.288 ± 0.012	U/μg of total protein	NO	0.1191
MDA	0.005 ± 0.0003	0.006 ± 0.0002	0.007 ± 0.0003 (**)	nmol/μg of total protein	YES	0.0086
Catalase	0.062 ± 0.003	0.061 ± 0.002	0.070 ± 0.004	U/μg of total protein	NO	0.1821
ATP	6.104 ± 0.476	6.045 ± 0.575	7.239 ± 0.224	nmol/μg of total protein	NO	0.1882

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
