# Peer review of "Evaluation of the Adverse Effects of Chronic Exposure to Donepezil (An Acetylcholinesterase Inhibitor) in Adult Zebrafish by Behavioral and Biochemical Assessments"

_biomolecules, 2020, doi:10.3390/biom10091340_

Round 1

Reviewer 1 Report

The present study is generally well-written and well-illustrated. Its strengths include a wide range of phenotypes covered, robust effects reported, and a nuanced characterization of DPZ effects. However, the overall study design is only targeting chronic DPZ treatment, and misses shorter-term treatment designs, as well as the respective literature. For example, a recent study by Giacomini et al 2020 (in press, J of Psychopharm), has tested 24-h DPZ in adult zebrafish, yielding anxiogenic-like effects of this drug, paralleled by analyses of AChE activity in the brain. The present study reports opposite effects by a longer-term chronic DPZ treatment. Such difference in results for acute vs semi-acute vs chronic treatments is not uncommon for various CNS drugs. Accordingly, the discussion here needs to address this difference, and put the present study in the contract of prior literature.

The authors appreciated the reviewer’s comment. Unfortunately, the authors found difficulty in finding the recent study mentioned by the reviewer. Thus, if it is possible and appropriate, the full information about the recent study, such as the title of the study or preprint paper, provided by the reviewer will be very helpful for the authors. However, regarding the difference between acute vs chronic treatments that might happen in results, the possible explanation based on the previous study was added to the manuscript, specifically in the Discussion part. The difference might happen since this study found that donepezil induced changes in a number of neurotransmitters, such as dopamine, serotonin, and norepinephrine, in some specific regions, including the rat dorsal hippocampus, in a time-dependent manner.

Minor comments:

Abstract is very long and hard to follow - please cut by 50%

The authors strongly agreed with your suggestion. Thus, the authors had tried our best to make the revised abstract shorter as the reviewer suggested by removing some unnecessary sentences or information.

Title: Must reflect chronic DPZ treatment aspect of the study

Thank you very much for the correction. As the reviewer’s suggestion, the title of the present study had changed, so now it is “Evaluation of the Adverse Effects of Chronic Exposure of Donepezil (An Acetylcholinesterase Inhibitor) in Adult Zebrafish by Behavioral and Biochemical Assessments“.

Methods - please describe better the test battery

Thank you very much for the suggestion. Therefore, as the reviewer suggested, a more detailed description of all of the battery of zebrafish behavioral tests was added to the manuscript, specifically in the Experimental Methodology part.

Also, provide a diagram illustrating the overall study design

The authors strongly agreed with the suggestion. Thus, the schematic diagram to show the overall study design was added to the appendix data section (Figure A2).

Please provide statistical power calculation to ensure that the study is not underpowered

Thank you for the suggestion. Indeed, the statistical power is important to validate research and it is necessary to determine the size effect. However, the sample size applied in the current experiment was based on several prior studies, especially in the effect of some CNS drugs in zebrafish behavior as mentioned below.

de Abreu, M. S., Giacomini, A. C., dos Santos, B. E., Genario, R., Marchiori, N. I., da Rosa, L. G., & Kalueff, A. V. (2019). Effects of lidocaine on adult zebrafish behavior and brain acetylcholinesterase following peripheral and systemic administration. Neuroscience letters692, 181-186.

Genario, R., Giacomini, A. C., de Abreu, M. S., Marcon, L., Demin, K. A., & Kalueff, A. V. (2020). Sex differences in adult zebrafish anxiolytic-like responses to diazepam and melatonin. Neuroscience letters714, 134548.

Kolesnikova, T. O., Khatsko, S. L., Eltsov, O. S., Shevyrin, V. A., & Kalueff, A. V. (2019). When fish take a bath: Psychopharmacological characterization of the effects of a synthetic cathinone bath salt ‘flakka’on adult zebrafish. Neurotoxicology and teratology73, 15-21.

Volgin, A. D., Yakovlev, O. A., Demin, K. A., Alekseeva, P. A., & Kalueff, A. V. (2019). Acute behavioral effects of deliriant hallucinogens atropine and scopolamine in adult zebrafish. Behavioural Brain Research359, 274-280.

Statistical analyses: How was data normality tested, to qualify for parametric vs non-parametric testing 

The authors appreciated the question. In terms of the distribution of the data, all of the data that were tested by non-parametric analysis had their distribution tested by several tests, including Anderson-Darling, D’Agostino & Pearson, Shapiro-Wilk, and Kolmogorov-Smirnov tests, first. Since there was no behavioral endpoint that showed normal distribution in all of the normality tests, the non-parametric analysis was applied in these data. All of the important information regarding the data normality had been added to the manuscript, specifically in the Statistical Analysis part.

Reviewer 2 Report

The manuscript studied the effects of donepezil, a drug approved for the treatment of Alzheimer's disease, on the behavioral and biochemical parameters of zebrafish. Although the approach is interesting, the manuscript has relevant limitations mainly concerning methodological and statistical aspects. As the methodological aspects were described, it is very difficult to replicate the experimental protocol. I raised some aspects below:

1) The description of the materials and methods has some limitations. Important aspects regarding animal care and randomization were not adequately described.  I suggest that the authors review the ARRIVE guidelines (PLoS Biol. 2010 Jun 29;8(6):e1000412) to better describe the methodology.

The authors appreciated the reviewer’s effort to enhance the materials and methods part. As the reviewer suggested, the authors had read the reference provided by the reviewer and tried our best to update the materials and methods part according to the reference. These changes were including animal care and randomization mentioned by the reviewer.

2) A major point is regarding randomization and blinding. Masca et al. (2015) and Gerlai (2018), examining the problems of reproducibility in biomedical research, state that one of the main reasons for irreproducible research is the lack of blinding/masking. The author did not describe that randomization was performed to allocate the animals to the “treatment” groups. What was the method used for random allocation in the treatment groups? Were experimenters blind to treatment? Were data analysts blind to treatment? These questions need to be addressed and clearly stated in the methods section. 

Thank you for reminding the authors about this important matter. Actually, the blind analysis was applied in the current experiment by giving codes to the samples so that as the experimenters and data analysts conducted the experiment and analyzed the data, respectively, they were unaware of prior treatments. In addition, the simple random allocation method was used in all of the fish groups. All of this information were added to this revised manuscript, specifically in the Experimental Methodology part.

3) How was the sample size calculated? Was it based on a pilot experiment to determine effect sizes? How were the calculations performed? 

Thank you for the questions. The sample size applied in the current experiment was based on several prior studies, especially in the effect of some CNS drugs in zebrafish behavior as mentioned below. Meanwhile, the calculations were performed by using the GraphPad Prism software as it was mentioned in the Experimental Methodology part.

de Abreu, M. S., Giacomini, A. C., dos Santos, B. E., Genario, R., Marchiori, N. I., da Rosa, L. G., & Kalueff, A. V. (2019). Effects of lidocaine on adult zebrafish behavior and brain acetylcholinesterase following peripheral and systemic administration. Neuroscience letters692, 181-186.

Genario, R., Giacomini, A. C., de Abreu, M. S., Marcon, L., Demin, K. A., & Kalueff, A. V. (2020). Sex differences in adult zebrafish anxiolytic-like responses to diazepam and melatonin. Neuroscience letters714, 134548.

Kolesnikova, T. O., Khatsko, S. L., Eltsov, O. S., Shevyrin, V. A., & Kalueff, A. V. (2019). When fish take a bath: Psychopharmacological characterization of the effects of a synthetic cathinone bath salt ‘flakka’on adult zebrafish. Neurotoxicology and teratology73, 15-21.

Volgin, A. D., Yakovlev, O. A., Demin, K. A., Alekseeva, P. A., & Kalueff, A. V. (2019). Acute behavioral effects of deliriant hallucinogens atropine and scopolamine in adult zebrafish. Behavioural Brain Research359, 274-280.

4) It is not clear for how many days the animals were exposed to the treatment?

Thank you for the correction. Totally, the zebrafish were exposed to the treatment for 21 days. However, two sections of experiments were conducted in the current study. In the first section, the effect of a high concentration of donepezil was studied in the healthy zebrafish after 21 days of incubation. Since there was no significant benefit found after the treatment, another experiment was carried out in order to observe the adverse effect of donepezil in this high concentration, especially on zebrafish behaviors. In addition, to further examine its effect in a low concentration, another treatment group, which was 1 ppm, was also tested in this experiment. This experiment consisted of novel tank, mirror biting, predator avoidance, social interaction, and shoaling tests, which were conducted in day 14±1, color preference test, which was carried out in day 17, circadian rhythm locomotor activity test, which was conducted in day 19, and biochemical assays, which were carried out in day 21. All of this crucial information was added to the revised manuscript, specifically in the Experimental Methodology part (Figure A2).

5) For each treatment, were the animals kept in only one tank? If so, this could be a problem. I suggest that at least two aquariums be used for each treatment, and also, the tank effect should be considered in the statistical analysis. Did the authors verify the potential effect of the tank? It means, how many tanks were used for each treatment?

The authors appreciated the question. In the current experiment, the animals of each treatment were kept in a single tank with 4L of either normal water, 1 ppm, or 2.5 ppm of donepezil. This method was applied since several prior studies, as mentioned below; found that this stocking density of zebrafish for the long term will help to maintain low-stress levels. Moreover, regarding the usage of a single tank, this method was also based on several prior studies that also conducted a similar concept with the current study as mentioned below and also to reduce the donepezil usage amount. However, if this explanation is not reasonable enough, it will be very helpful if several literature showing the study regarding the tank effect in the incubation process of zebrafish are provided, thus, this can be the limitation of the current study and should be addressed in the manuscript. In addition, the information regarding the zebrafish rearing was added to this revised manuscript, specifically in the Experimental Methodology part.

Aleström, P., D’Angelo, L., Midtlyng, P. J., Schorderet, D. F., Schulte-Merker, S., Sohm, F., & Warner, S. (2019). Zebrafish: Housing and husbandry recommendations. Laboratory Animals, 0023677219869037.

Castranova, D., Lawton, A., Lawrence, C., Baumann, D. P., Best, J., Coscolla, J., ... & Wilson, C. (2011). The effect of stocking densities on reproductive performance in laboratory zebrafish (Danio rerio). Zebrafish8(3), 141-146.

Lidster, K., Readman, G. D., Prescott, M. J., & Owen, S. F. (2017). International survey on the use and welfare of zebrafish Danio rerio in research. Journal of fish biology90(5), 1891-1905.

Rabbane, M. G., Rahman, M. M., & Hossain, M. A. (2016). Effects of stocking density on growth of zebrafish (Danio rerio, Hamilton, 1822). Bangladesh Journal of Zoology44(2), 209-218.

Dlugos, C. A., & Rabin, R. A. (2003). Ethanol effects on three strains of zebrafish: model system for genetic investigations. Pharmacology Biochemistry and Behavior74(2), 471-480.

Grossman, L., Stewart, A., Gaikwad, S., Utterback, E., Wu, N., DiLeo, J., ... & Kalueff, A. V. (2011). Effects of piracetam on behavior and memory in adult zebrafish. Brain Research Bulletin85(1-2), 58-63.

Meshalkina, D. A., Kysil, E. V., Antonova, K. A., Demin, K. A., Kolesnikova, T. O., Khatsko, S. L., ... & Kalueff, A. V. (2018). The effects of chronic amitriptyline on zebrafish behavior and monoamine neurochemistry. Neurochemical research43(6), 1191-1199.

6) Did the author verify the distribution of the data before choosing the statistical test? The authors should improve the description of the results. Statistical data must be provided (F, degrees of freedom, p values). Moreover, the number of animals in each test is lacking or is inconsistent along with the figures (this information should be presented in the figure legends).

The authors strongly agreed with the reviewer regarding the improvement of the description of the results. Therefore, the statistical data had provided for each test in the results section. Furthermore, in terms of the distribution of the data, all of the data that were tested by non-parametric analysis had their distribution tested by several tests first. Since there was no behavioral endpoint that showed normal distribution in all of the normality tests, the non-parametric analysis was applied in these data. In addition, the number of animals in each test already consistent with the figures. This argument can be clarified simply since in most of the graphs, each individual value was displayed. However, for the other graphs that did not show the individual values, the authors had checked again the number of animals and its consistency with the figures. All of the important information regarding the reviewer’s point had been added to the manuscript.

7) Post hoc tests should only be carried out when a significant interaction effect is obtained (F1 x F2) – when this is not the case, only the main effects should be shown in the figures, preferably with symbols other than asterisks (please check Nieuwenhuis et al., Erroneous analyses of interactions in neuroscience: a problem of significance. Nat Neurosci. DOI: 10.1038/nn.2886). This means the interpretation of most results must be carefully reconsidered.

Thank you for mentioning this statistic issue. Indeed, post-hoc are used when an analysis of variance is significant and conducted in order to find the significant differences between control and the treatment groups, which could not be shown by the analysis of variance only. Thus, as the reviewer’s suggestion, the P value of the main effect of each endpoint was added and symbolled by “#”. For the circadian rhythm locomotor activity test, it was shown separately in Table A1 since the authors felt that the space in the Figure was not sufficient.

8) How were the animals fed? How many times a day? What was the interval between eating and behavioral experiments? How many hours were the animals fasting during the experiments?

Thank you for addressing this important matter. The animals were fed twice daily, at 09:00 and 17:00, with fresh brine shrimp and commercial dry food. The interval between eating and behavioral experiments was around 1-2 hours. The animals were not fasting during the experiment to avoid the external factor affecting their behavior, such as starving. The information regarding the feeding process of the fish was added to this revised manuscript, specifically in the Experimental Methodology part.

9) Were the experiments replicated?

The authors appreciated the question. In the current study, the experiments were not replicated, since replication is of little value if there were no doubts about the original experiment setup.

10) What is the relationship between the doses used in this study and those used in the clinic? Has a relationship been established between these concentrations and plasma and/or central concentrations found in patients using this drug? As it is presented I have the impression that they are random doses without a relationship with the clinic.

Thank you for the question. In this study, the doses used were based on several prior studies in several animal models, especially rats, as mentioned below. Donepezil in a similar concentration with the current study enhanced the survival of newborn neurons in adult rats and indicated positive effects of donepezil treatment on specific cognitive performances in cholinergically depleted rats. In addition, donepezil in this concentration can be employed clinically for the treatment of cognitive deficits after chemotherapy in a rat model. Therefore, the present study aimed to confirm whether the same effect of donepezil in a similar concentration was also found in zebrafish. In addition, this important information regarding the concentration of donepezil used in the current study was added to the manuscript, specifically in the Experimental Methodology part.

Creeley, C. E., Wozniak, D. F., Nardi, A., Farber, N. B., & Olney, J. W. (2008). Donepezil markedly potentiates memantine neurotoxicity in the adult rat brain. Neurobiology of aging29(2), 153-167.

Cutuli, D., Foti, F., Mandolesi, L., De Bartolo, P., Gelfo, F., Federico, F., & Petrosini, L. (2009). Cognitive performances of cholinergically depleted rats following chronic donepezil administration. Journal of Alzheimer's Disease17(1), 161-176.

Jiang, L., Wang, Y., Su, L., Ren, H., Wang, C., Chen, J., & Fu, X. (2019). Donepezil Attenuates Obesity-Associated Oxidative Stress and Central Inflammation and Improves Memory Deficit in Mice Fed a High-Fat Diet. Dementia and Geriatric Cognitive Disorders48(3-4), 154-163.

Kotani, S., Yamauchi, T., Teramoto, T., & Ogura, H. (2008). Donepezil, an acetylcholinesterase inhibitor, enhances adult hippocampal neurogenesis. Chemico-biological interactions175(1-3), 227-230.

Lim, I., Joung, H. Y., Yu, A. R., Shim, I., & Kim, J. S. (2016). PET evidence of the effect of donepezil on cognitive performance in an animal model of chemobrain. BioMed research international2016.

11) I did not understand why in figure 1 both doses of donepezil were not used. Please explain.

Thank you for the suggestion. In Figure 1, only one concentration (2.5 ppm) was used because, in this experiment, the authors only aimed to study the positive effect of donepezil in healthy zebrafish’s memory performances. However, since in this high concentration its effect was not observed, the authors decided not to use a lower concentration. This consideration was taken since there was a tendency that donepezil in a lower concentration would not show any positive effect in zebrafish’s memory performances if the higher concentration did not show its effect. In addition, the authors intended to use animals as few as possible to fix the 3R guideline of animal welfare. This explanation was added to the manuscript.

12) Figure 3: Was the tank position random between each animal?

The authors appreciated the question. In the current behavioral tests, especially mirror biting and predator avoidance tests as mentioned by the reviewer (Figure 3), the tank position was random between each animal.

13) Appendix: The number of animals is varying in each figure. The authors should explain the inconsistencies between the figures.

Thank you for the correction. Indeed, there was a variation of n number used in several behavior tests. This variation was occurred due to the death of the zebrafish that was caused by an unidentified reason. This important explanation was added to the manuscript, specifically in the Experimental Methodology part.

14) I am not sure that the increase in time in the upper zone of the tank is an anxiolytic effect as the animals showed motor changes. In this case, I suggest that the authors present the data of total distance and number of crossings between the zones of the tank.

The authors appreciated the reviewer’s comment. However, the increase in time in the upper zone of the tank is already a well-known paradigm to show an anxiolytic effect of a chemical on the animals, especially zebrafish. Some references that mentioned about this matter were mentioned below. Furthermore, regarding the reviewer’s suggestion about the presentation of the data of total distance and number of crossings between the zones of the tank, the authors felt that it would be redundant to some behavioral endpoints that already mentioned in Figure 2. The total distance was already represented by the average speed since the average speed is the total distance divided by the total time, which was the same for every time interval. Further, the number of crossing between the zones was already represented by the number of entries to the top endpoint.

Ansai, S., Hosokawa, H., Maegawa, S., & Kinoshita, M. (2016). Chronic fluoxetine treatment induces anxiolytic responses and altered social behaviors in medaka, Oryzias latipes. Behavioural brain research303, 126-136.

Genario, R., de Abreu, M. S., Giacomini, A. C., Demin, K. A., & Kalueff, A. V. (2020). Sex differences in behavior and neuropharmacology of zebrafish. European Journal of Neuroscience52(1), 2586-2603.

Hamilton, T. J., Morrill, A., Lucas, K., Gallup, J., Harris, M., Healey, M., ... & Tresguerres, M. (2017). Establishing zebrafish as a model to study the anxiolytic effects of scopolamine. Scientific reports7(1), 1-9.

Kellner, M., Porseryd, T., Hallgren, S., Porsch-Hällström, I., Hansen, S. H., & Olsén, K. H. (2016). Waterborne citalopram has anxiolytic effects and increases locomotor activity in the three-spine stickleback (Gasterosteus aculeatus). Aquatic Toxicology173, 19-28.

Sackerman, J., Donegan, J. J., Cunningham, C. S., Nguyen, N. N., Lawless, K., Long, A., ... & Gould, G. G. (2010). Zebrafish behavior in novel environments: effects of acute exposure to anxiolytic compounds and choice of Danio rerio line. International journal of comparative psychology/ISCP; sponsored by the International Society for Comparative Psychology and the University of Calabria23(1), 43.

15) The authors need to improve graph resolution.

Thank you for the suggestion. The authors had tried our best to improve graph resolution.

Round 2

Reviewer 1 Report

The authors provided sufficient and careful responses to most of my technical comments. However, two items from the previous review remain to be addressed further.

1) The study on behavioral effects of donepezil in zebrafish must critically draw upon the only previous work that recently examined this drug CNS effects in zebrafish. This recent study by Giacomini et al 2020 (in press, J of Psychopharm), has tested 24-h DPZ in adult zebrafish, yielding anxiogenic-like effects of this drug, paralleled by analyses of AChE activity in the brain.  The authors noted difficulty in finding the recent study mentioned by the reviewer. I am attaching here the details of this pertinent study DOI: 10.13140/RG.2.2.31376.64006 Giacomini et al. An acetylcholinesterase (AChE) inhibitor donepezil increases anxiety and cortisol levels in adult zebrafish. Accordingly, the discussion needs to put the present study in the context of prior literature. 

Thank you very much for providing more detailed information about the previous work in the acute exposure effect of DPZ in adult zebrafish. The authors already read the report thoroughly and carefully. As the reviewer suggested, the discussion regarding the result of the prior study and the current study was added to the manuscript, especially in the discussion part. The possible reason caused the difference in the present study results and previous work results were described in the manuscript.

2) This reviewer asked to provide statistical power calculation to ensure and justify that the study is not under-powered. The authors replied citing several other studies that have similar n's. However, this is not sufficient to address the question about this particular study. Please provide statistical power analysis for this study and justify in the methods section the selected sample size. Please consult expert biostatisticians, if necessary, to ensure the correctness of such analyses.

The authors appreciated the suggestion and understood that by citing several other studies that have similar n’s is not enough to address the question about this particular study. Thus, the authors tried their best to provide the statistical power calculation to ensure that the study is not underpowered. From the statistical power calculation with 90% confidence interval and a margin of error of 7 units, it was found that the sample size of 25 is needed, which was similar to n number applied in the current study. All of the explanation regarding the statistical power calculation was added to the manuscript, specifically in the Experimental Methodology part and supplementary data. The authors hoped that this revision is in agreement with the reviewer’s suggestion. If it is not, it will be very helpful for the authors if the reviewer showed some of the prior studies in this field that provided a statistical power calculation.

Reviewer 2 Report

The manuscript is acceptable in the present form.

Round 3

Reviewer 1 Report

This is a resubmission of the paper that was carefully modified in line with the expert reviewers' concerns. I have no further comments.